# Timing and Frequency of Daily Energy Intake in Adults with Prediabetes and Overweight or Obesity and Their Associations with Body Fat

**DOI:** 10.3390/nu12113484

**Published:** 2020-11-13

**Authors:** Christina Sonne Mogensen, Kristine Færch, Lea Bruhn, Hanan Amadid, Inge Tetens, Jonas Salling Quist, Kim Katrine Bjerring Clemmensen

**Affiliations:** 1Steno Diabetes Center Copenhagen, DK-2820 Gentofte, Denmark; Kristine.faerch@regionh.dk (K.F.); lea.bruhn.nielsen@regionh.dk (L.B.); hanan.amadid.01@regionh.dk (H.A.); jonas.salling.quist@regionh.dk (J.S.Q.); kim.katrine.bjerring.clemmensen.01@regionh.dk (K.K.B.C.); 2Department of Biomedical Sciences, University of Copenhagen, DK-2200 Copenhagen N, Denmark; 3Department of Nutrition, Exercise and Sports Copenhagen University, DK-2000 Frederiksberg, Denmark; ite@nexs.ku.dk

**Keywords:** dietary intake, eating patterns, obesity: body composition, prediabetes

## Abstract

Knowledge on how energy intake and macronutrients are distributed during the day and the role of daily eating patterns in body composition among adults with overweight/obesity and prediabetes is lacking. Therefore, we evaluated the diurnal dietary intake and studied the associations of daily eating patterns with body fat percentage. A total of 119 adults with prediabetes were included (mean (SD) HbA_1c_ 41 (2.3) mmol/mol, BMI 31.5 (5.0) kg/m^2^, age 57.8 (9.3) years, 44% men). Information on dietary intake was obtained from self-reported food records for three consecutive days. All foods and beverages (except water) were registered with information on time of ingestion. Body fat was measured by dual-energy X-ray absorptiometry. A total of 60.5% of the participants reported a daily eating window of 12 or more hours/day, and almost half of the daily total energy intake was reported in the evening. In analyses adjusted for age, gender, and total daily energy intake, having the first daily energy intake one hour later was associated with slightly higher body fat percentage (0.64% per hour, 95% CI: 0.28; 1.01; *p* < 0.001), whereas higher meal frequency was associated with slightly lower body fat percentage (0.49% per extra daily meal, 95% CI: −0.81; −0.18; *p* = 0.002). Prospective studies are warranted to address the clinical implications of daily eating patterns on body fat and cardiometabolic health.

## 1. Introduction

The prevalence of obesity has increased worldwide since the 1980s [1] and in 2016, 39% of adults were overweight and 13% were obese [2]. Obesity is associated with an increased risk of morbidity and mortality [1] and weight loss is associated with a decreased risk of several conditions, including type 2 diabetes, cardiovascular disease, and non-alcoholic fatty liver disease [3]. Therefore, weight loss has high priority in the prevention and treatment of these conditions.

The fundamental cause of overweight is an imbalance between energy consumed and energy expended. Restricting energy consumption or increasing energy expenditure by e.g., physical activity may induce weight loss [1], but these lifestyle changes are often difficult to maintain in the long term [4]. Therefore, new strategies that can be easily implemented and maintained are needed to combat the current obesity epidemic.

Most dietary guidelines focus on total energy intake and macronutrient composition [5,6,7], but these do not include advice on the timing of energy intake and distribution of macronutrients during the day. Strategies that exploit timing of energy intake as a means to achieve weight loss or improving metabolic health have been the subject of considerable public and academic interest. One weight loss strategy, termed “time-restricted eating”, limits the daily eating window (interval between the first energy intake in the morning and the last energy intake in the evening) without any dietary restriction and appears to reduce body weight in individuals with overweight and obesity [8,9,10]. Another strategy for weight loss is eating more frequently, which, in a randomized controlled trial, appeared to reduce hunger in healthy, lean, young males and thus reduced energy intake and body weight [11]. In contrast, however, a cross-sectional study in young and old participants showed that lower meal frequency was associated with lower BMI [12]. Moreover, eating meals later during the day seems to adversely influence the success of weight loss [13] and cardiometabolic risk in overweight adults [14].

In general, knowledge on when people eat and how the macronutrients are distributed during the day is lacking because previous dietary studies have mainly focused on the average energy intake as well as macro- and micronutrient composition. Furthermore, it is unknown how the timing of daily energy intake is associated with body composition. Therefore, we evaluated the dietary intake among adults with prediabetes and overweight or obesity and studied the associations of daily eating patterns, including a daily eating window (time-interval from the first energy intake in the morning to the last energy intake in the evening), meal frequency (number of meals/day), and time of first intake and time of last intake with body fat percentage in individuals with prediabetes and overweight or obesity. We hypothesized that, after adjusting for total energy intake, later meal timing, lower meal frequency, and a longer eating window would be associated with higher body fat percentage.

## 2. Materials and Methods

The present study is a post-hoc analysis based on baseline data from the PRE-D Trial, which is described in detail elsewhere [15]. The study was approved by the Ethics Committee of the Capital Region (H-15011398) and the Danish Medicines Agency (EudraCT number: 2015-001552-30) and was conducted in accordance with the Declaration of Helsinki. All participants received written and oral information about the study and signed an informed consent before taking part in the study.

### 2.1. Participants

Altogether, 120 participants were included. Advertisements in newspapers, online media and recruitment of relatives to patients with diabetes at the Steno Diabetes Center Copenhagen, Gentofte, Denmark were used as recruitment strategies. Written information about the study was sent to 772 potential participants. A total of 404 participants were interested in the study and participated in a screening visit. Participants who were eligible after screening were included. Inclusion criteria were prediabetes defined as HbA_1c_ 39–47 mmol/mol (5.7–6.4%), age ≥ 30 to ≤70 years, and BMI ≥ 25 kg/m^2^. For a full list of inclusion and exclusion criteria see the PRE-D Trial protocol article [16].

For the present analysis, one participant with shift work was excluded due to the potential impact of shift work on eating patterns, leaving 119 participants for analysis.

### 2.2. Clinical Examination

The participants came into the research facility at the Steno Diabetes Center Copenhagen in the morning after an overnight fast (>8 h) and without exercising for 48 h. Height was measured three times at the screening visit by a stadiometer (SECA, Hamburg, Germany), and the average height was used, and body weight was measured at all visits using a weight scale (TANITA, BWB-620A, Vienna, Austria) with participants wearing light clothes and no shoes. Body fat was measured using dual-energy X-ray absorptiometry (DEXA) scan (HOLOGIC, Santax medico, Discovery QRD Series, Marlborough, MA, USA). Body fat percentage was calculated as total body fat mass (kg) divided by total body weight (kg) measured by weight scale multiplied by 100 [17].

### 2.3. Dietary Intake

After the baseline visit, participants were instructed to fill in a dietary record (paper and pen) for the following three days. The participants were instructed to register all foods and beverages (except water) either as a serving or in grams, as well as the time of ingestion. Participants were provided with a digital kitchen scale (Soehnle Industrial Solutions GmbH, Backnang, Germany) with a load capacity of up to 5.0 kg with precision down to 1 g/0.1 oz. After the participants had filled in the dietary records, the dietary records were sent back to the clinic by mail. All participants were advised to follow the Danish national dietary guidelines [18,19] before leaving the research facility.

The dietary records were entered into Diætist Net Pro (Sweden) [20], a dietary software program which is based on data from the food databases: Fødevaredatabanken v7.01, Fabrikanter 2018-08-13, Livsmedelsverket 2017-12-15(v2), and Fineli 2018-02-28. Inputs were made in grams and nutrient intake was calculated via linkage to the food databases. Time of ingestion was recorded. Energy intake (i.e., all energy-containing food/beverages) was divided into the following meals: breakfast, lunch, dinner, and all other eating occasions labeled as snacks [14]. Whether the meals were categorized as breakfast, lunch, dinner, or snacks was defined based on food type and time of the day. We defined a meal as intake of any kind of energy-containing food or beverage. Meals were recorded as separate entities if the energy intakes were minimum 15 min apart [14].

To separate the first energy intake on one day from the last energy intake on the night before, the day was set to start at 4:00 am. This cut-point was chosen because no food intake was registered in the period from 2:00 to 4:00 during the night in our study, and the percentage of energy intake was lowest at 4 am in a study of youth with overweight [21]. Moreover, 4:00 am has previously been used to separate meal intake over two days [8].

### 2.4. Classification of Valid Reporters

To determine the proportion of potential mis-reporters of energy intake, the Goldberg cut-off methodology was used [22]. Basal metabolic rate (BMR) was calculated using the equation by Henry (2005) [23], which includes sex, age, height and weight. Based on the three-day period of dietary registration, a lower 95% CI cut-off point of 1.00 and an upper 95% CI cut-off point of 2.40 was chosen [22]. Thus, if a participant’s EI:BMR ratio for the three-day period was below 1.00 or higher than 2.40, the participant was defined as a mis-reporter. In the calculation, physical activity level (PAL), defined as EI:BMR, was set to 1.55, which is the World Health Organization (WHO) value for sedentary or light activity lifestyles which fitted the present study population [22,24].

### 2.5. Statistical Analyses

The characteristics of the participants are presented with means and standard deviations (SD). Data on daily eating patterns are presented as the mean of the three-day dietary registration period.

To describe daily eating patterns, bar plots and polar plots were used. To determine the associations of daily eating patterns, including the length of daily eating window (hours/day), meal frequency (meals/day), timing of meals (time of day of last energy intake before 4 am (hours), and first energy intake after 4 am (hours)) with body fat percentage, linear regression analyses were used. The analyses were performed for all 119 participants and separately for participants with acceptable reported energy intake according to the Goldberg methodology. Two linear regression models were used: one unadjusted and one adjusted for age, sex, and energy intake. All data were analyzed using R for Windows version 3.6.0. Statistical significance was determined as a two-sided *p* < 0.05.

## 3. Results

### 3.1. Characteristics

The characteristics of the participants are presented in Table 1. Because no examinations were performed during the weekend, Thursday and Friday were over-represented while Monday and Tuesday were under-represented. A total of 81 participants out of the 119 (68%) were acceptable reporters according to the Goldberg methodology, and 32% of the participants were under-reporters for both men and women. No participants over-reported their daily energy intake. Similar results were found in all participants and acceptable reporters alone and the associations of daily eating patterns and body fat percentage in all participants and in acceptable reporters alone were similar (see Appendix A).

### 3.2. Daily Eating Patterns

As illustrated in Figure 1a,b, the mean daily eating window varied largely among the participants. The mean (SD) daily eating window in all participants was 12.3 (1.8) h (Table 1). A total of 60.5% of the participants had a daily eating window of 12 or more hours/day, and 14.3% of the participants had a daily eating window of 14 or more hours/day. Only 9.2% had a daily eating window of less than 10 h/day.

The meals were distributed over a large part of the day for both women and men (Figure 1c). Participants consumed on average (SD) 5.6 (1.5) meals/day (5.6 (1.3) in women and 5.7 (1.8) in men).

### 3.3. Diurnal Variation in Intake of Energy and Macronutrients

The proportion of the daily total energy intake is shown in Figure 2. In total, 18% of the total energy was consumed in the time period from 04.00 to 09.59; 35% from 10.00 to 15.59; 45% from 16.00 to 21.59; and 3% from 22.00 to 03.59. The highest proportion of total daily energy intake per hour was reported between 18.00 and 18.59, where 14% of daily energy intake was reported (Figure 2).

Figure 3 shows the time of day and the proportions of macronutrients reported for each hour. In the time period from 04.00 to 09.59 the reported average hourly carbohydrate intake was 59%, whereas 19% came from protein and 25% came from fat. In the time period from 10.00 to 15.59, 50% of the reported hourly energy intake came from carbohydrate, 16% came from protein, and 34% came from fat. From 16.00 to 21.59, 42% came from carbohydrate, 17% came from protein, and 35% came from fat; and in the time period from 22.00 to 03.59, 50% of the average hourly energy intake came from carbohydrate, 10% from protein and 34% came from fat.

### 3.4. Relationship between Daily Eating Patterns and Body Fat Percentage

The daily eating window was not associated with body fat percentage in either the unadjusted model (*p* = 0.30) or the adjusted model (*p* = 0.184) (Figure 4). However, a higher meal frequency was associated with a lower body fat in the unadjusted model with −0.66% body fat per extra meal (95% CI: −1.17; −0.15; *p* = 0.01) and in the adjusted model with −0.49% body fat per extra meal (95% CI: −0.81; −0.18; *p* = 0.002) (Figure 4). Furthermore, a later intake of the first meal in the day was positively associated with body fat percentage in both the unadjusted model (1.17% (95% CI: 0.58; 1.77) per hour; *p* < 0.001) and the adjusted model (0.64% (95% CI: 0.28; 1.01) per hour; *p* < 0.001). Time of last meal was not associated with body fat percentage in the unadjusted model (*p* = 0.15) or in the model adjusted for age, sex, and total daily energy intake (*p* = 0.165) (Figure 4). No association was found between time of first intake and time of last intake adjusted for age and sex (*p* = 0.560).

## 4. Discussion

The present cross-sectional study is to our knowledge the first study that has evaluated time specific daily eating patterns in adults with prediabetes. The average daily eating window was ~12 h/day with only ~9% reporting a daily eating window of less than 10 h/day. Despite small effect sizes, we found that eating the first meal later in the day was associated with higher body fat percentage, whereas a higher meal frequency was associated with a slightly lower body fat percentage in overweight and obese adults with prediabetes. We observed no association between daily eating window and fat percentage.

The mean (SD) daily eating window of 12.3 (1.8) h in the present study is in line with a cross-sectional study that found a mean daily eating window of 12.2 (0.06) hours in 14,854 American adults (age ≥ 20 years old) using a 24-h recall [25]. A cross-sectional study with a 30-day protocol to describe circadian behaviors found no association between daily eating window and body fat percentage in 110 participants of mixed ethnicity, aged 18–22 years [26]. Additionally, the daily eating window was not associated with BMI in another cross-sectional study [8]. However, previous short term intervention studies have found that restricting the daily eating window without restricting energy intake, i.e., time-restricted eating, leads to weight loss [8,10,27,28,29].

We found that a higher meal frequency was associated with a slightly lower body fat percentage. This is consistent with our hypothesis and with a previous cross-sectional study using four 24-h recalls including 2696 adults (BMI: mean (SD) 28.3 ± 4.9 kg/m^2^) [30]. In that study, those with six or more eating occasions per day had a lower mean BMI compared with participants with fewer than four occasions per day (27.3 vs. 29.0 kg/m^2^) [30]. Moreover, a longitudinal study in female adolescents found that a low meal frequency, assessed by two three-day dietary records, predicted a higher BMI and waist circumference 10 years later [31]. However, previous RCTs and cross-sectional studies have been inconsistent as to whether an increase in meal frequency results in increased or decreased energy intake [30,32].

In our study, the average time of the first meal intake was at 08:23 a.m. while the average time of the last meal was at 08:25 p.m. (Table 1). This is in line with an American cross-sectional survey from 2009–2014, including 15,341 adults that found an average time of first meal at 08.08 a.m. and an average time of last meal at 08:18 p.m. [25]. Few studies have investigated sex differences in the timing of eating. Our data showed that men had their first meal at 08:04 a.m. and women at 08:22 a.m., which is in line with the findings from a previous study (08:03 (0:04) a.m. vs. 08:13 (0:04) a.m.) [25]. Timing of the last meal in our study corresponds well with the observations in the aforementioned study (08:22 p.m. in men and 08:13 p.m. in women) [25].

The mechanisms linking meal frequency and later meal timing to regulation of energy balance and body fat percentage are not well established. A study including healthy men with a mean (SD) age of 22.9 (4.2) years and BMI of 32.1 (2.8) kg/m^2^ found that eating more frequent and smaller meals compared with one larger meal during the day, lead to greater control of satiety scores, measured by visual analogue scales (VAS), with a subsequent reduction in energy intake at an ad libitum meal [11]. Furthermore, a 13-day laboratory study including 12 healthy, normal weight adults found that appetite measured by VAS peaks in the evening [33]. Results from clinical studies show that eating meals at later times may decrease satiety hormones and increase subjective appetite [34,35,36]. An intervention study including 93 overweight and obese women (BMI 32.4 (1.8) kg/m^2^) with metabolic syndrome were randomized into two isocaloric (~1400 kcal) weight loss intervention groups; a breakfast group (700 kcal breakfast, 500 kcal lunch, 200 kcal dinner) or a dinner group (200 kcal breakfast, 500 kcal lunch, 700 kcal dinner) for 12 weeks [35]. The group who consumed more energy at breakfast showed a greater weight loss, higher waist circumference reduction, improved appetite scores (VAS) during the day, and had an greater reduction in fasting glucose, insulin, and ghrelin levels compared with the group who ingested more energy at dinner [35]. Taken together, individuals who eat a larger proportion of the total energy intake earlier in the day seem to reduce the daily total energy intake [34,37] and have a greater effectiveness when attempting weight loss [13,30,35,38]. In our study, no associations were found between time of first meal and time of last meal, adjusted for age and gender.

Systematic measurement error due to misreporting (especially low energy reporting) is an acknowledged problem in dietary assessment methods. Underreporting in particular is a limitation in food diaries, and may derive from (1) changes in true intake as a function of recording or being observed; (2) lack of awareness concerning food items and amounts consumed; and/or (3) reluctance to disclose food or amounts eaten [39]. However, the Goldberg equation is one of the most widely used methods for identifying under-reporters [22,40]. The proportion of under-reporters in our study was 32%, which is comparable to what has been seen in other studies using food records [41]. A major strength of the present study is the detailed food records including the time of intake of meals and beverages. Weighed dietary records are considered the gold standard when examining dietary intake under free-living conditions [42]. Another strength of this study is the measurement of body fat percentage using a DEXA scan for the assessment of body composition [17]. This method provides more reliable and accurate estimates of body fat percentage than other anthropometric measures, and it is well accepted that body fat percentage is superior to BMI in measuring adiposity [17].

The present study is cross-sectional and, therefore, causality on the relationship between timing of dietary intake and body fat cannot be determined. Another limitation is that the study population is relatively small and only included adults with prediabetes and overweight or obesity. Furthermore, many of the participants were retired and therefore ate their first meal slightly later in the day than people who go to work would do. These factors limit the generalizability of our findings to other populations. Because of the limited sample size and lack of information on e.g., genetics and chronotype, we did not adjust for all factors, which may be of relevance for the association between daily eating patterns and body fat percentage. Therefore, there is a risk of residual confounding. Furthermore, it cannot be excluded that registration of eating patterns of three continuous days may not be representative of a habitual dietary pattern. The reported energy intake is usually higher during weekend days compared to weekdays [42]; therefore, it would have been favorable to include both weekdays and weekend days for all participants, but for logistic reasons this was not possible. Based on a scientific statement from the American Heart Association, a meal must be separated by at least 15 min and provide 50 kcal to be considered unique [14]. In our assessment, the interval between meals was at least 15 min, but we did not use any criteria with regards to the energy content of a meal. All energy intakes have both physiological and metabolic consequences with an effect on energy balance; therefore, all foods and beverages containing energy were included in this study. Thus, it is possible that the eating frequency is different in the present study compared with studies using other definitions of a meal.

Diet restrictions often result in rapid weight loss, but weight maintenance is challenging, and barriers for maintenance include lack of knowledge and support, as well as confusion about quantity and type of recommended foods [43]. Our findings suggest that timing of energy intake during the day might be associated with body fat percentage. More studies are needed to clarify whether there is a potential for considering daily eating patterns in dietary recommendations in the prevention and treatment of overweight and obesity. By increasing our knowledge on the role of timing of food intake in human health, new feasible strategies and recommendations, which could be easier to follow, can be developed.

## Figures and Tables

**Figure 1 nutrients-12-03484-f001:**
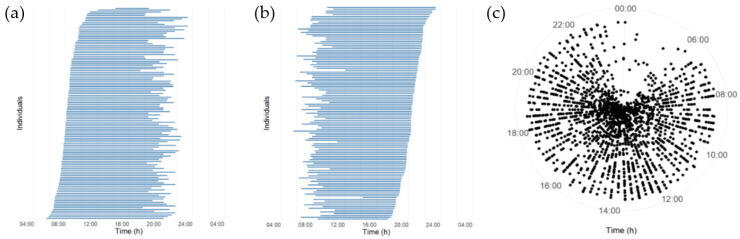
A schematic overview of eating patterns on an individual level. (**a**,**b**): Each horizontal bar shows the average length of the daily eating window for a three-day period for a participant. Each bar starts with the average clock-time of first energy intake (beginning of eating window) to the average clock-time of last intake (end of eating window). Daily eating window (hours) is shown in order of the latest (top) to the earliest (bottom) time of first intake (**a**) and last intake (**b**). (**c**) All registered meals of each individual are plotted against time of the day (radial axis) in each concentric circle (*n* = 119).

**Figure 2 nutrients-12-03484-f002:**
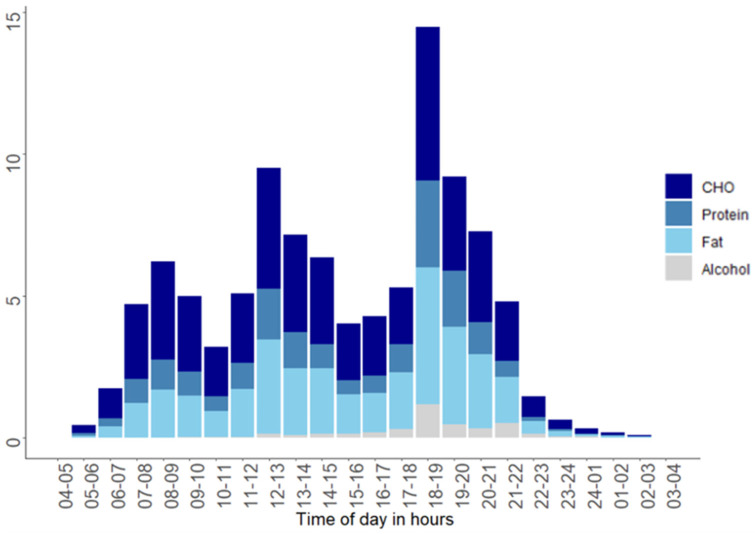
Mean proportion (%) of the daily energy intake per hour stratified by macronutrients. Each bar shows the average energy intake per hour of carbohydrate, protein, fat, and alcohol for the three-day period. CHO: carbohydrate, E%: Energy percentage, h: hours.

**Figure 3 nutrients-12-03484-f003:**
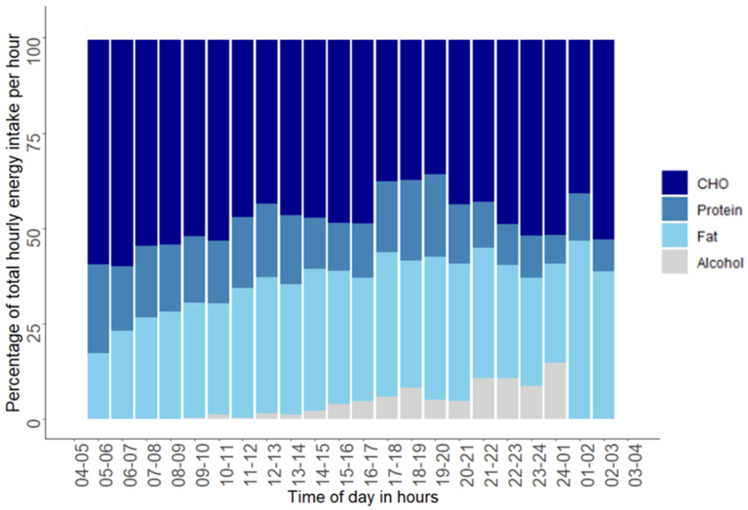
Energy percentage of total energy intake per hour for each macronutrient. Each bar represents macronutrient distribution in energy percentage of the average hourly energy intake. No energy intake was reported between 03:00 and 04:59 h.

**Figure 4 nutrients-12-03484-f004:**
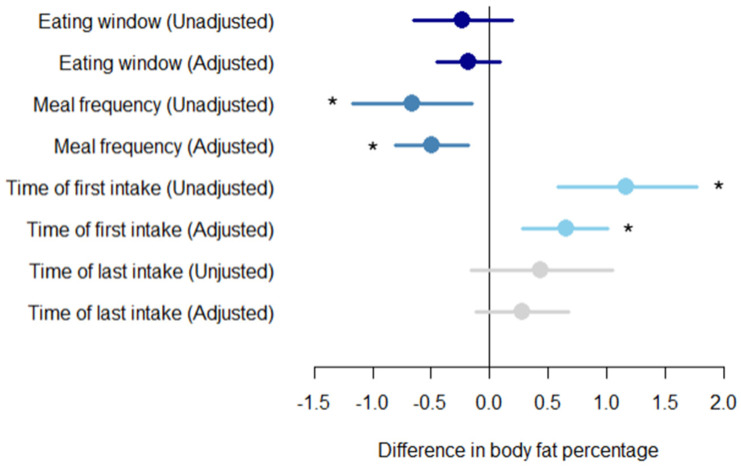
Daily eating patterns and body fat percentage. Associations between body fat percentage for each increment in exposures for different daily eating patterns including daily eating window, meal frequency, time of first intake and time of last intake. The analysis is unadjusted and adjusted for age, gender, and total daily energy intake. An increment in daily eating window is one hour. An increment in meal frequency is one meal. An increment in time of first intake and last intake is one hour. The circles represent effect sizes; extended lines show 95% confidence intervals. * *p*-value < 0.05.

**Table 1 nutrients-12-03484-t001:** Characteristics, dietary intake and eating patterns for the participants.

	All (*n* = 119)	Women (*n* = 66)	Men (*n* = 53)
Age (years)	57.8 (9.3)	57.9 (8.9)	57.7 (9.7)
Height (m)	1.71 (0.09)	1.65 (0.06)	1.80 (0.07)
Weight (kg)	93.3 (17.8)	86.0 (16.2)	102.3 (15.5)
BMI (kg/m^2^)	31.5 (5.0)	31.4 (5.8)	31.6 (3.9)
Body fat (%)	38.0 (7.4)	43.3 (4.5)	31.4 (4.4)
HbA_1c_ (mmol/mol)	41 (2)	41 (2)	41 (2)
**Dietary intake**			
Daily energy intake (kJ/day)	8246 (2578)	7402 (2301)	9299 (2523)
CHO (g)	226 (78)	205 (70)	252 (78)
CHO (E%)	46.6 %	47.1 %	46.1 %
Fat (g)	75 (31)	70 (31)	81 (30)
Fat (E%)	33.7 %	35.0 %	32.2 %
Protein (g)	85 (23)	77 (18)	96 (25)
Protein (E%)	17.5 %	17.7 %	17.6 %
Alcohol (g)	11 (18)	5 (7)	18 (23)
Alcohol (E%)	3.8 %	2.0 %	5.6 %
**Daily eating patterns**			
Daily eating window (hours)	12.3 (1.8)	12.2 (1.6)	12.3 (2.1)
Number of meals (meals/day)	5.6 (1.5)	5.6 (1.3)	5.7 (1.8)
Time of first intake (hours)	8.23 (1.3)	8.36 (1.1)	8.07 (1.4)
Time of last intake (hours)	20.41 (1.3)	20.46 (1.3)	20.35 (1.2)

Data are presented as mean (standard deviation). BMI: body mass index, Body fat (%): body fat percentage, CHO: carbohydrate.

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
