# Peer review of "Timing and Frequency of Daily Energy Intake in Adults with Prediabetes and Overweight or Obesity and Their Associations with Body Fat"

_nutrients, 2020, doi:10.3390/nu12113484_

Round 1

Reviewer 1 Report

In this study, daily eating patterns in adults with prediabetes and overweight or obesity in relation to body fat are studied. This is a highly relevant topic, and the paper is well written. Please find my specific comments below:

  1. The finding that having the first energy intake 1h later is associated with higher body fat percentage seems not in line with the hypothesis that a shorter time window is associated with a better cardiometabolic profile, and thus lower body fat. From the paper, it is not clear if those individuals who start later also have a later clock time of their last caloric intake? In addition, would it be possible investigate the time window between last caloric intake and bedtime? Please elaborate on this in the discussion of the paper.
  2. Please elaborate on sample size; does the number of 119 participants provide sufficient statistical power for analyses?
  3. In the title and abstract, ‘dietary patterns’ is too vague. It can also indicate ‘Mediterranean’ or ‘Nordic’ diets. Please specify.
  4. Please provide your definitions of breakfast, lunch and dinner (line 100).
  5. Are cut off points for time windows of morning, afternoon and evening (lines 156-158) literature-based? What is the motivation for these timeslots? How do other studies or countries define morning, afternoon or evening? In most countries, afternoon starts after noon, not already at 10 am. Why are clock times between 4 and 6 pm not included in afternoon or evening? (line 157). Please elaborate on this in the methods and discussion.
  6. The proportion of invalid reporters is quite high. How can this be explained? Is this proportion what can be expected?
  7. Does data collection by the DEXA in the PRE-D trial enable the researchers to specify between fat mass and lean mass, and between SAT, VAT and liver fat?
  8. With data collection from 3 food diaries, it would also be possible to study meal irregularity, which has also been shown to affect body weight.
  9. In statistical analyses I miss adjustment for important covariates such as dietary quality, education/SES, smoking, chronotype/social jetlag etc. Was information on these characteristics not collected? Please elaborate on this in the discussion as a limitation or add fully adjusted analyses to the paper
  10. The average first meal at ~8.30 am seems to be quite late. Is this in line with Danish National Survey of Diet?
  11. Results from this study only apply to individuals with prediabetes having overweight. In the discussion, authors can elaborate on any potential dietary advice people in DK receive upon diagnosis of prediabetes and how this may affect results.
  12. Authors need to mention the limitation of food diaries in that people tend to change their actual food intake while writing down what they consume, and how this may affect study findings.
  13. Definitions of a meal in line 262 could be moved to methods section.

Reviewer 2 Report

Summary: This study with Mogensen et al evaluates baseline data from the PRE-D Trial which consists of 120 prediabetic adults. The focus of the study revolves around how the distribution of macronutrients along with information on energy intake affects the body composition/body fat percentage of adults with prediabetic status.

The findings of this study are as follows:

  • Over 60.5% of participants reported a daily eating window of >12 hours/day
  • Almost half of the energy consumption was made during the evening
  • Having the first intake of energy be one hour later had associations with higher body fat
  • Having higher meal frequency was associated with lower body fat

Positives: This study utilizes actual human data, may have more translational relevance compared to studies using only animal models. The unique study which relates adult body fat in prediabetics to energy intake and macronutrient distribution. May provide a stepping stone to also gaining insight into cardiometabolic health.

Questions/Areas of Improvement:

  • The PRE-D Trial referenced by this paper [15] by FÆRCH et al, employed a study of 120 participants with 2 groups undergoing drug treatments: Dapagliflozin and Metformin, it is unclear if this study considered any potential effects from these participants and whether or not they were included/excluded or if there are potential effects of these drugs in affecting patient profile.
  • Overall, the paper seems to pursue an interesting idea but lack key information to have a strong convincing message. For example, time-restricted feeding has been shown to improve profiles of obese associated phenotypes but has no significant improvement in this study with no justification of why, just a citing of one paper which also found no beneficial effect. Despite the presented key figures, there is a seemingly small amount of data and the study of participants was only done in the span of 3 days, more time may be needed to get more accurate results.
  • It is confusing why the authors would expect higher meal frequencies to result in higher body fat stated in the introduction hypothesis. Two references were given one which stated that higher meal frequency is used as a strategy for weight loss [11] and another which states that young and old individuals had lower body fat with a lower frequency of meals [12]. Despite the two contrasting points the majority/consensus is that higher meal frequency results in an increase in metabolism and lower body fat, why did the authors hypothesize the opposite?
  • No consideration was seemingly made about genetics and how much of a role genetics may play in altering body fat or whether all patients have a similar genetic background with no family history of being prediabetes.
  • Also, another consideration is the consistency of meal timings, how often does person A eat at the time “a”, time “b” and time “c”? Or do they always have different times in which they eat? Eating inconsistently affects the metabolic clock. Just related to that, the categorizations of “afternoon”, “evening” and “late-night evening” are confusing because 16.00 – 17.59 doesn’t belong to any category and 22.00-23.59 falls into two categories. More explanation will be needed to justify the categorization.
  • As the dietary food content can be a big contributor to body fat, it will be also interesting to have different sub-groups based on the total amount of fat intake, and then re-evaluate the association between diet patterns and body within the population that consume similar daily fat intake.

More Minor points/Grammatically errors:

  • There are multiple grammar and spelling errors throughout in the manuscript and these need to be addressed.

Reviewer 3 Report

Thank you for the opportunity to review this manuscript  regarding clinical implications of daily eating  patterns on body fat and cardiometabolic health. The study is well performed. 

Howewer, I am less enthusiastic aboud the conclusions of the authors and the should modify and clearly supported by the presented data. 

I have questions, a few comments for the authors: 

Introduction: 

The introduction provides suffienct background and include all relevant references about association with overweight and obesity individuals instead of  adults with prediabetes , that is the other basic aim study. So, these references could be added.

Materials and methods:

Line 104 : The cut point used is based on only one references?? this reference is not a review .... could you add some references to estabilihed this cut off?

Line 105: Indicate the acronym SD in the text 

Results : 

Table 1: The table must be better aligned to the center of text 

table 1: To align the value about protein (17.7 %) to the center 

Discussion: 

Line 223-224: You say that there are sex differences in the timing of eating....are there significant differences?? From the results obtained it doesn not seem . 

As I already mentioned in the introduction about association in prediabetes patients,  also in the discussion,  references are very limited.

How  do the authors explain the above results? Wha are the conclusions? The conclusions should reflect upon the aims - whether they were achieved or not??  

"

Round 2

Reviewer 2 Report

All the questions/concerns raised in the previous version have been adequately addressed and I don't have any questions. This should be suitable for publication now. best wishes,  

Author Response

Response to Reviewer 2 comments

Thank you for reviewing the manuscript.

 All the questions/concerns raised in the previous version have been adequately addressed and I don't have any questions. This should be suitable for publication now. best wishes,  

Moderate English changes required:

Response: I have found some spelling errors in line 115, 119, and 146 these have been corrected.

Reviewer 3 Report

I appreciate your response to my reviewer ’ comments. 

I have only some questions.

1) As I have already reccomended , if you answered to my two comments that,   this study focuses on meal patterns in relation to body fat percentage and not glucose metabolis , why you foculised the aim on prediabetes adults also in the title ???

2) Line 104 : The cut point used is based on only one references?? this reference is not a review .... could you add some references to estabilihed this cut off?

You've answered... satisfactorily but not to my main question ?? Are there othere references about this cut off meal point used?? or only one??
